# Effect of Probiotics on Host-Microbial Crosstalk: A Review on Strategies to Combat Diversified Strain of Coronavirus

**Susrita Sahoo [1], Swati Mohapatra [2,3,*], Swayam prava Dalai [2] , Namrata Misra [1,4] and Mrutyunjay Suar [1,4]**

1   School of Biotechnology, Kalinga Institute of Industrial Technology (KIIT-DU), Bhubaneswar 751024, India;
    ss.sahoo8976@gmail.com (S.S.); misranamrata8@gmail.com (N.M.); mrutyunjaysuar80@gmail.com (M.S.)
2   Department of Infection Biology, School of Medicine, Wonkwang University, Iksan 54538, Korea;
    swayampravadalai@gmail.com
3   Amity Institute of Microbial Technology, Amity University-Utter Pradesh, Noida 201313, India
4   KIIT-Technology Business Incubator (KIIT-TBI), Kalinga Institute of Industrial Technology (KIIT),
    Deemed to Be University, Bhubaneswar 751024, India
*   Correspondence: swati1990@wku.ac.kr

**Abstract:** The scare of the ongoing coronavirus disease 2019 (COVID-19) pandemic, caused by the severe acute respiratory syndrome coronavirus 2 (SARS-CoV-2), does not seem to fade away, while there is a constant emergence of novel deadly variants including Alpha, Beta, Gamma, Delta and Omicron. Until now, it has claimed approximately 276,436,619 infections, and the number of deaths surpluses to 5,374,744 all over the world. While saving the life has been a priority during the ongoing SARS-CoV-2 pandemic, the post-infection healing and getting back to normalcy has been undermined. Improving general health conditions and immunity with nutritional adequacy is currently of precedence for the government as well as frontline health workers to prevent and assuage infections. Exploring the role of probiotics and prebiotics in managing the after-effects of a viral outbreak could be of great significance, considering the emergence of new variants every now and then. To enhance human immunity, the recent evidence on the connection between gut microbiota and the broad spectrum of the clinical COVID-19 disease is the reason to look at the benefits of probiotics in improving health conditions. This review aims to sketch out the prospective role of probiotics and prebiotics in improving the standard of health in common people.

**Keywords:** probiotics; COVID-19; microbiota; viruses; infection; therapeutics

## 1. Introduction

Various acute respiratory tract infections caused by viruses, including respiratory syncytial virus, enterovirus, pneumonia-causing viruses, adenovirus and influenza virus, are the main causes of debility and death worldwide [1]. The main causative agent for these respiratory tract infections (RTIs) are DNA/RNA viruses. However, the RTIs associated with RNA viruses are more virulent in comparison to those that are caused by DNA viruses [2]. Specifically, coronaviruses belong to a highly significant re-emerging RNA virus family, causing serious life-threatening respiratory infections [3]. Ever since the onset of the infectious coronavirus disease (popularly known as COVID-19) in Wuhan city of China, the pandemic has increased rapidly in 57 countries, with over 276 million COVID-19 cases and over 5.37 million deaths reported as of 27 December 2021. Additionally, imposing several socio-economic, proper feedback strategies and rigid public health measures globally, involving social distancing, mask wearing, personal hygiene, quarantines, and lockdowns, the number of infections and death due to SARS-CoV-2 virus are constantly rising [4].

Given the continuous evolution of the virus that leads towards SARS-CoV-2, WHO, in collaboration with researchers, national authorities, institutions and expert networks, monitored the emergence of variants that posed an increased risk to global public health and prompted the characterization as Variants of Interest (VOIs) and Variants of Concern

(VOCs). Currently, using comparative assessment strategies, WHO labeled five variants as VOCs Alpha (United Kingdom, September-2020), Beta (South Africa, May-2020), Gamma (Brazil, November-2020), Delta (India, October-2020)and Omicron (Multiple countries, November-2021), while Lambda (Peru, December-2020) and Mu (Colombia, January-2021) were labeled as VOIs.

With the continuous boost from WHO, current ongoing research trends and developmental attempts are completely focused on developing effective therapy to counter the novel virus [5]. In this direction, anticoagulants, convalescent plasma, Hydroxychloroquine, Remdesivir, Vasodilators, non-steroidal anti-inflammatory drugs, monoclonal antibodies and Lopinavir/Ritonavir are in distinct phases of trials, research, or approvals. However, none of the above treatments are completely effective against the virus [6–8]. In the absence of potent and efficacious vaccines and medicines, the virus is severely transforming and exhibits symptomatic, pre-symptomatic and asymptomatic forms in the affected population. Both asymptomatic and pre-symptomatic exemplifications are certainly one of the principal reasons for the pandemic [9]. Moreover, WHO released an assessment that this novel disease might persist in staying with the global population for a prolonged period. Therefore, proper investment and constant preparedness in public health and other resources are required for supervising the spread and morbidity caused by SARS-CoV-2.

The novel COVID-19 exhibits wide diversity in disease severity, spanning from minor and ill-defined common cold-like symptoms to pneumonia, and then can lead to life-threatening complications such as acute respiratory distress syndrome (ARDS) and multiple organ failure [10]. The proliferation and transmission of SARS-CoV-2 are caused through respiratory droplets; however, Ng et al. reported that the gut could also play a major role in the pathogenesis of COVID-19 [11]. Moreover, it was also reported that some coronaviruses, including the present SARS-CoV-2, could infect enterocytes, thus plating as a potential reservoir for virus proliferation [12]. Altogether, few clinical reports have revealed that gastrointestinal symptoms are conventional in COVID-19-infected patients, and in few cases, it leads to disease severity [13,14]. In addition, current pandemic control measures and practices to manage pandemics implement long-term effects on the human microbiome across the world, given the imposition of fleeing endemic areas, physical distancing, scapegoating of certain groups, mask-wearing, personal hygiene, quarantines, and lockdowns that influence overall microbial loss and inability for reinoculation. Rapid depletion and reduction of microbes over generations may lead to the extinction of microbial species ancestrally associated with humans; species may be permanently lost from the microbial pool unless reinoculation from other sources occurs [15,16]. In this context, various reports have suggested probiotic strains as promising therapeutics to enhance human immunity, thus inhibiting pathogens colonization and further minimizing the incidence and intensity of the infections. Moreover, little clinical evidence also illustrated the significance of probiotics in preventing viral and bacterial infections, including RTIs, sepsis, and gastroenteritis [17].

Keeping in view the enormous health and economic burden, repurposing the usage of natural compounds such as probiotics and prebiotics can be an effectual therapeutic approach in blocking and/or reducing SARS-CoV-2 severity. In this review, we describe the existing curative and preventive trial studies focused on the usage of probiotics and prebiotics to combat viral infections. Moreover, the possible application of probiotics bacteria as a prophylactic approach against COVID-19 is also outlined in the present study.

## 2. SARS-CoV-2 and COVID-19

Morphologically, the enveloped SARS-CoV-2 viruses harbor a single-stranded non-segmented positive-sense ribonucleic acid (RNA) genome. The novel SARS-CoV-2 harbors completely diverse virus from formerly identified coronaviruses, i.e., Middle East respiratory syndrome coronavirus (MERS-CoV) and SARS-CoV [18]. For that reason, the previously available flu or antiviral therapeutics is ineffectual against it.

The majority of studies revealed that glycoproteins are engaged in binding to the host and consequent virus–host membrane fusion to produce the pathogenesis of the SARS-CoV-2 [19]. Structurally, the four major glycoproteins involving the membrane protein (M), small envelope protein (E), spike protein (S), and nucleocapsid (N) proteinform the general structure of coronavirus [20] (Figure 1). The surface of virus harbors *N*-linked glycosylated trimeric S of 150 kDa that directs *N*-terminal signals towards the host endoplasmic reticulum. The M (25–30 kDa) with a higher *C*-terminal endodomain and lesser *N*-terminal glycosylated ectodomain is responsible for the shape of the virion [7]. Several vertebrate reservoirs, including humans, camels, dogs, masked palm civets, bats, cats and mice, are the potent hosts for coronaviruses [21]. Previous studies have revealed that COVID-19 was initially harbored by bats and then subsequently transmitted to humans through the infection of wild animals; however, the consequent spread of the virus into the human population occurred through human-to-human transmission [22,23]. The approximate time period between the introduction of the SARS-CoV-2 and its symptom onset in the host species, the incubation period, is 1–14 days [24]. Moreover, the standardization and estimation of the incubation period for COVID-19 may vary depending on the host's age, the genetics of the individuals, the environmental conditions [25], the pathogenicity of the virus or the long-term use of specific treatment such as glucocorticoids [26,27]. The usage of glucocorticoids might cause atypical infections and can also increase the incubation period. In addition, variability was also observed in the clinical manifestations of COVID-19 infections, varying from no or minimal symptoms to severe viral pneumonia with failure of respiratory organs and even death [28]. Earlier reports have revealed that asymptomatic or pre-symptomatic COVID-19 patients can play as promising resources for disease transmission [29,30]. Some familiar symptoms of COVID-19 include cough, fever, sore throat, fatigue, shortness of breath, aches, myalgia and headache [31]. Other associated symptoms reported include discoloration of fingers or toes, a rash on the skin or conjunctivitis, and some gastrointestinal symptoms such as vomiting, nausea, diarrhea and abdominal pain [32]. Person-to-person dissemination of COVID-19 infections was possible via cough- or sneeze-respiratory droplets released from the mouths/noses of SARS-CoV-2-infected patients [33]. Moreover, direct contact with the contaminated surfaces is an alternative means of COVID-19 transmission in humans [34]. With the onset of the disease, the infected person faces trouble in breathing, and simultaneously, the disease leads to severe respiratory tract infection and chronic inflammation [35].

Depending on the Cryo-EM structural investigation of S, it was revealed that the main reason for the rapid spread of SARS-CoV-2 is the S protein, having a 10–20 times higher affinity to human angiotensin-converting enzyme 2 (ACE2) receptor, in comparison to the previously emerged SARS-CoV [36]. Moreover, Zou et al. recently revealed that human organs, including epithelial cells from alveolar (lung) and enterocytes belonging to small intestines, are possible targets of the deadlySARS-CoV-2 virus [37]. Very recently, Guan et al. detected the SARS-CoV-2 viruses in contaminated human stools, recommending the chances of fecal–to-oral transmission [38]. Supporting this report, Holshue et al. further corroborated in a few patients from US and China that SARS-CoV-2 viruses are able to grow and proliferate in both digestive and respiratory tracts [39]. Moreover, Wu et al. revealed that the fecal samples of some of the COVID-19 recovered individuals were identified to be positive for the RNA of SARS-CoV-2, even though their respiratory samples were tested negative [40]. In addition, various current research has confirmed that COVID-19 disease has adversely affected the physiology and anatomy of the gastrointestinal tract for an extended period thus is damaging the gut microbiota [41,42]. Conticini et al. performed a post-mortem investigation on a patient who had died because of COVID-19 complications involving liver, lung and heart tissue, which revealed that serious damage occurred to the lungs with edema and desquamation [43]. Additionally, some of the patients infected with SARS-CoV-2 exhibited intestinal and microbial dysbiosis with reduced probiotic species instance, i.e., *Bifidobacterium* and *Lactobacillus*, signifying the necessity to look into the gastrointestinal and nutritional function of all patients [44–46]. These studies clearly indicate that the development

of opportunistic pathogens and simultaneous lessening of beneficial bacteria in GIT can be directly correlated with the severity of SARS-CoV-2 infections.

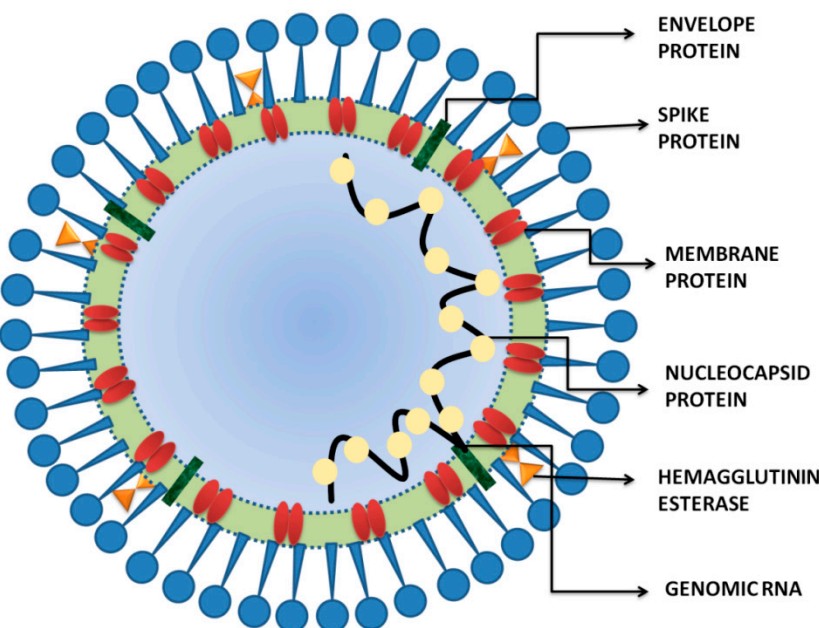

**Figure 1.** The schematic structure of SARS-CoV-2. The viral surface proteins, spike, envelope and membrane proteins. The single-stranded positive-sense viral RNA is associated with the nucleocapsid protein.

## 3. Discussion

The FAO/WHO definition of a probiotic is "live microorganisms which when administered in adequate amounts confer a health benefit on the host". [17,47]. Probiotics serveas an enormous metabolic advantage since they play a major role in host immunity by enhancing both specific and non-specific immune system [48]. Probiotics belonging to the genera *Lactobacillaceae*, *Leuconostocaceae* and *Bifidobacterium* [49] have been commonly utilized for their wide variety of benefits to the health by significantly reducing the loss of body weight, pathological symptoms, and viral loading [50–53]. The most commonly used probiotic microorganisms against pathogens include *Lactococcuslactis*, *Streptococcus thermophilus*, *Lactobacillus helveticus*, *Lactobacillus acidophilus*, *Lactobacillus delbrueckii* spp. *bulgaricus*, *Lactobacillus gallinarum*, *Lactobacillus amylovorus*, *Levilacto bacillusbrevis*, *Lactobacillus crispatus*, *Lactobacillus plantarum*, *Lactobacillus crispatus*, *Latilactobacillus curvatus*, *Limosilactobacillus fermentum*, *Lactobacillus johnsonii*, *Lacticaseibacillus paracasei* subsp. *paracasei*, *Lactobacillus delbrueckii* subsp. *lactis*, *Limosilacto bacillusreuteri*, *Lactobacillu scellobiosus*, *Lacticaseibacillus rhamnosus*, *Bifidobacterium laterosporum*, *Leuconostocmesenteroides*, *Pediococcus acidilactici*, *Pediococcus pentosaceus*, *Bifidobacterium adolescentis*, *Bifidobacterium animalis*, *Bifidobacterium breve*, *Bifidobacterium bifidum*, *Bifidobacterium infantis*, *Bifidobacteriumessensis*, *Bifidobacterium thermophilum*, *Bifidobacterium cereus*, *Propionibacterium acidipropionici*, *B. longumlongum*, *Alkalihalobacillus alcalophilus*, *Propionibacterium thoenii*, *Propionibacteriumjensenii*, *Propionibacterium freudenreichii*, *Enterococcus faecalis*, *Enterococcus faecium*, *Bacillus Clausii*, *Bacillus subtilis*, *Bacillus coagulans*, *Sporolactobacillus inulinus* and *Escherichia coli* [54–56]. Traditionally, probiotics were administered as fermented food and determined to enhance health and nutrition by repairing the microbial balance in the host GIT [57]. In recent years, various research has illustrated the function of probiotics in controlling immune responses and a wide variety of conditions, exclusively targeting the infections caused by viruses, in various clinical trials and animal models (Figure 2) [58]. Reports also suggested that dietary meals containing probiotics and fiber supplementation are required to prevent adverse effects of viral infections and maintain a stable immune response system within the host [59–61]. Most importantly, probiotic bacterial species, viz., *B. bifidum* [62], *Bacillus subtilis* [63], *L.*

*plantarum* [64,65] and *L. casei* [66] were previously reported to play a major role in providing a protective immune response against respiratory tract viral infection in experimental animal models. Moreover, Lehtoranta et al. reviewed that the interventions of probiotics lead in the reduction of viral load in the lungs by easing clinical symptoms, improving health conditions, and increasing survival rates [67–72].

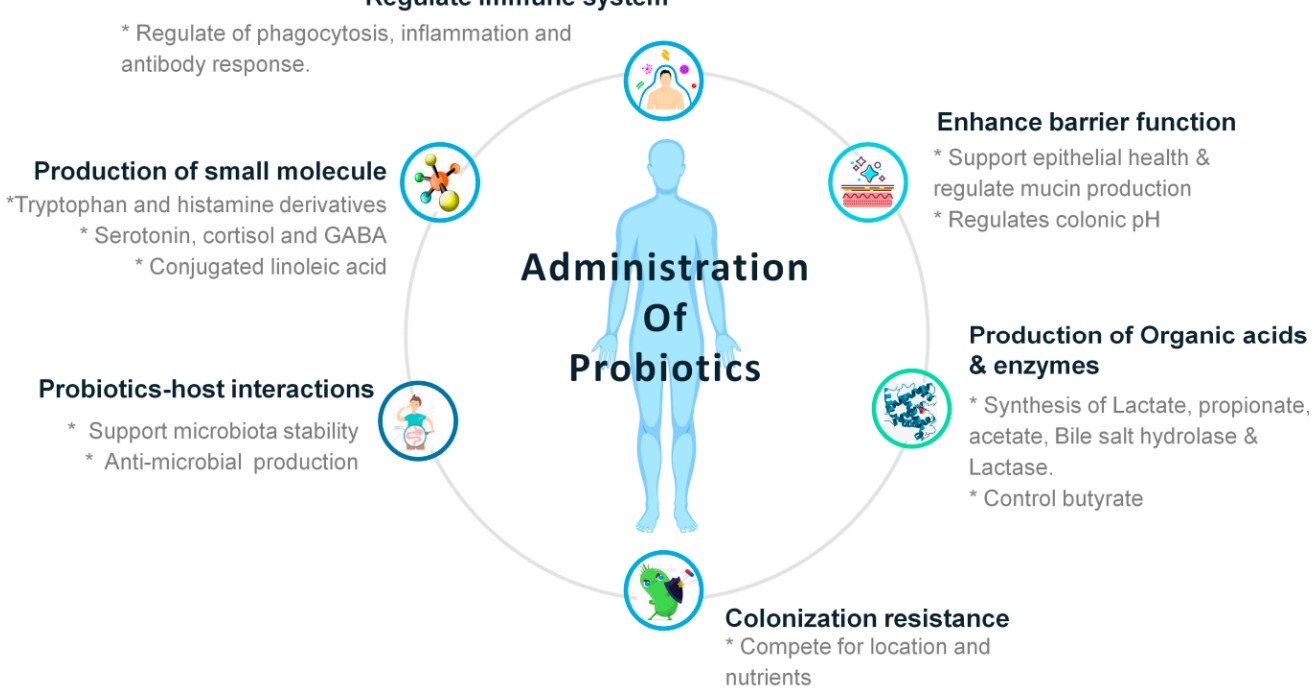

**Figure 2.** The schematic diagram representing the benefits of probiotics after administration into human health.

In addition to the GIT, probiotics also colonize at distant mucosal sites, including the lungs, and also enhance the systemic immune responses [50]. The secreted proteins of probiotics accelerate the production of Antigen-Presenting Cell (APC) that leads to secretion of various interleukins, such asIL10, IL12, IL17 and TNF-$\alpha$, interferon-$\alpha$ to eradicate foreign and allergic particles that trigger adaptive immunity. Probiotics provide two different immunomodulatory reactions: one is the immunostimulatory effect that activates IL-12 production, induces NK, Th, and Th2 cells, and acts against infection and allergy; and another type is the immunoregulatory effect, which induces IL-10 and Treg cell activation by Th2, DCs, B cells and monocytes for adaptive immunity of the host [7] (Figure 3).

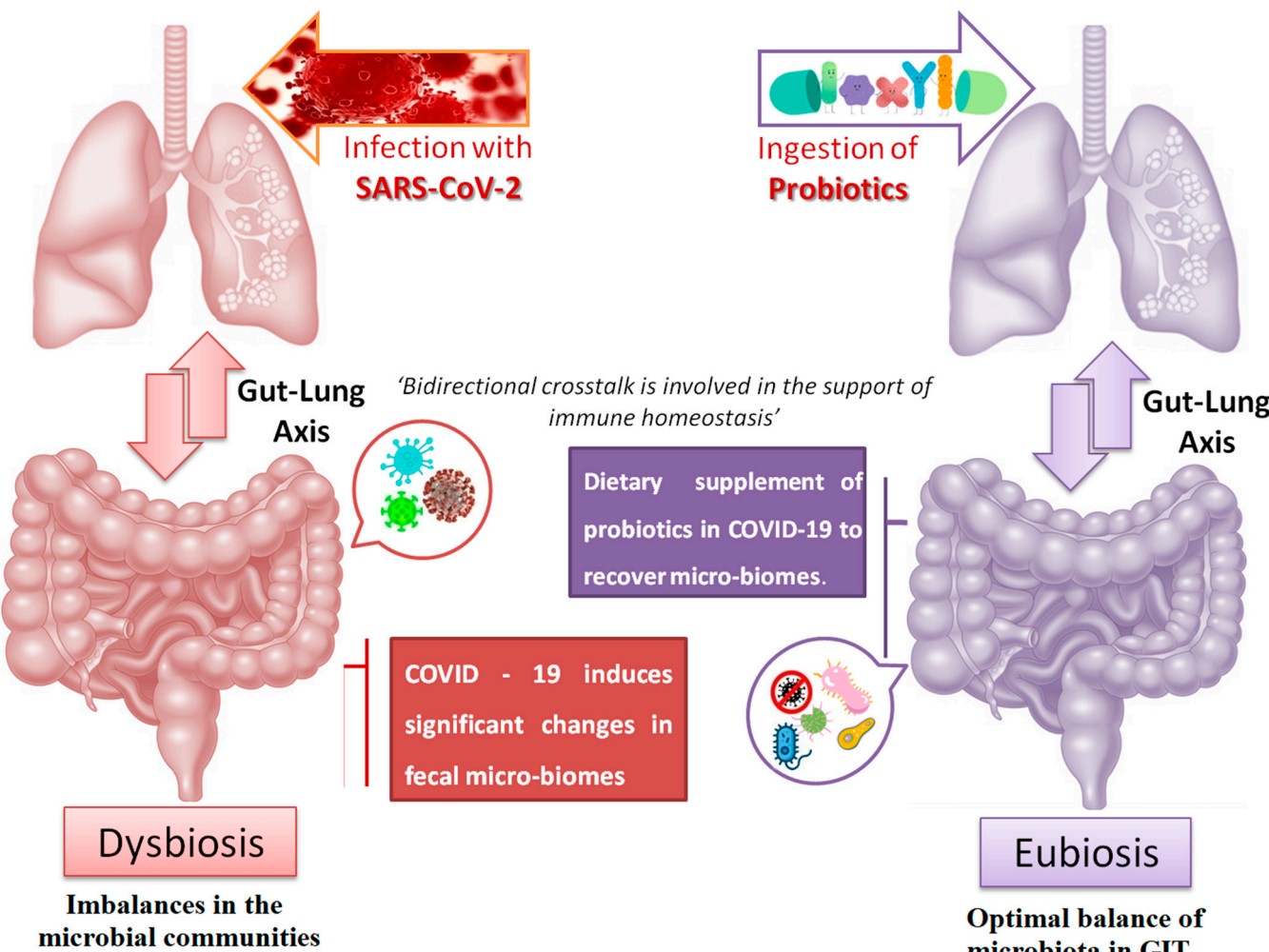

**Figure 3.** Schematic representation of the bidirectional crosstalk between the gut–lung axis.

## 4. COVID-19 Affecting the Gut–Lung Axis Crosstalk

Human microbiota plays a major role in the development and regulation of host metabolism, immune system, brain function and maintenance of a robust and resilient healthy homeostasis [16]. The gut and lung are among the sections in the human body that harbors microbiota; besides, the lung hosts a smaller amount of microbiota in comparison to gut. This bidirectional crosstalk between gut and lung is involved in supporting the immune homeostasis [73]. Moreover, previous reports have revealed that dysbiosis of microbiota from the gut is directly affected by various respiratory pathological conditions (Figure 4) [74,75]. Most importantly, metabolites and microbial components belonging to the gut viz. short-chain fatty acids and lipopolysaccharides are also engaged in the bidirectional communication of the gut–lung axis. In addition to the commonly reported respiratory symptoms such as cough, fever and severe respiratory syndrome caused by the infection COVID-19, research has reported that few COVID-19-infected patients also depicted GIT symptoms such as GI bleeding, loss of appetite, abdominal pain, diarrhea, nausea, and vomiting [76]. In a two-hospital cohort study, Yeoh et al. revealed that patients infected with COVID-19 were depleted in gut bacteria with known immunomodulatory potential even after disease resolution. Further, these complications lead to the increased concentrations of inflammatory cytokines and blood markers such as C reactive protein, lactate dehydrogenase, aspartate aminotransferase and gamma-glutamyl transferase [77]. In addition, various COVID-19 risk-reductions measures such as vaccination, masking, physical distancing, intensive hygiene and antibiotics negatively affect microbial diversity and accelerate microbiota loss [15].

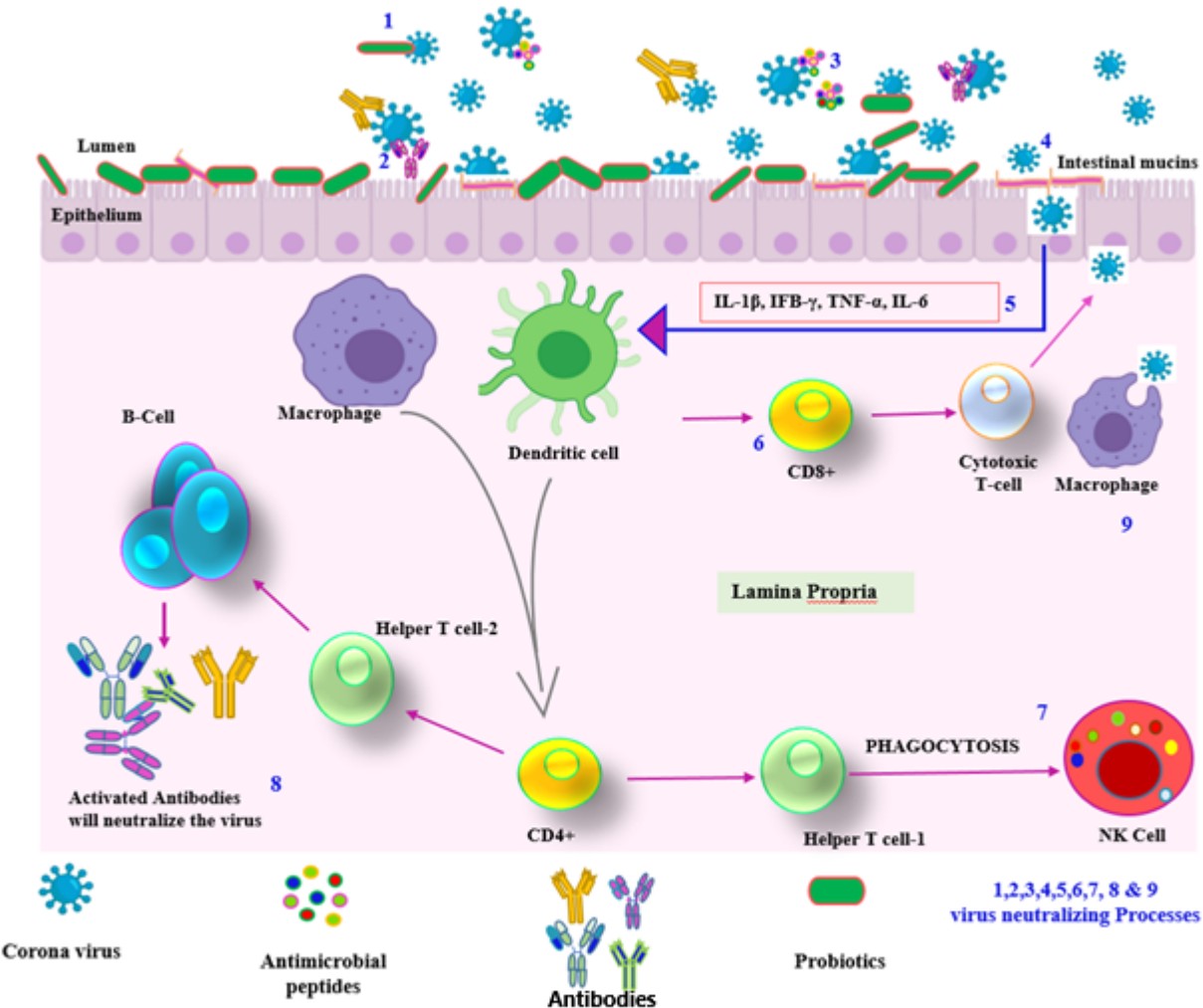

**Figure 4.** Schematic diagram representing the immune-modulatory agents regulated by probiotics.

Notably, Chiba et al. reported that SARS-CoV-2-infected patients with GIT symptoms including diarrhea suffered severe respiratory disorders when compared to patients without GIT symptoms [78]. However, very little information is offered regarding the consequence of lung microbiota on the microbiome of GIT. Moreover, few studies have revealed that acute lung injury regulates the dysbiosis of the lung that directly affects the blood-mediated modulation of the gut microbiota [79,80]. It was observed that the microbiota population was disrupted in cases of pulmonary allergy [78]. Keeping this view, we can hypothesize that COVID-19 infection can stimulate disruption of lung microbiota, which further regulates the microbiota from GIT, resulting in various GIT symptoms. Later, a few reports unveiled that the GIT symptoms developed in COVID infected patients might be attributed to the damaged tissues and organs caused by the immune responses [81–84]. In addition to the lung, it is reported that ACE2 is also identified in GIT, and direct colonization of the gut ACE2 receptors through the ingestion of the virus is probably liable for the GIT symptoms in connection with COVID-19. Furthermore, malfunction of apoptosis pathways in the intestine due to infections in the respiratory tract [85] is another projected elucidation for COVID-19-related GIT symptoms. Moreover, it can be believed that COVID-19-related GIT symptoms may be the consequence because gut and respiratory tracts have a common embryonic origin, and hence they share a similar structure and interact in a similar way in both pathological and physiological conditions [86]. Collectively, all the above mechanisms can assist researchers in understanding the GIT disturbances associated with COVID-19.

## 5. Supporting Evidence of Usage of Probiotics to Combat COVID-19

Despite several probable medications to treat the newly emerging SARS-CoV-2, there is always a constant increment in the number of death cases. Moreover, it has been observed that with intake of an optimized amount of probiotic supplements, most people are withstanding COVID-19 on account of booted immunity. The implication of probiotic strain, specifically *Bifidobacteria* and *Lactobacilli*, uplifted the health benefits and a significant stimulation towards recovery [50].

Significantly, various studies have depicted that the changes in the lung microbial community also influence the composition of gut microbiota due to a bidirectional relationship. Therefore, any type of infection in the lung can directly affect the intestinal bacterial environment. Hence, in order to maintain the healthy intestinal microbiota and cure the infections in the lungs, consumption of a significant amount of probiotic supplements will not only help the intestine but also stimulate the secretion of metabolites that can cure the contaminationin the lungs [24]. Previous clinical studies depicted the profit of probiotics towards nullifying the influenza virus present in the respiratory tract and reinforcing the lungs' immune system [17]. The administration of probiotics into the body enhances immunity- and anti-inflammatory cytokines, helping to clear the viral infection by minimizing the cell damage in the lungs [17]. Moreover, a clinical study has proven the exclusive impact through meta-analysis, where they have demonstrated external supplementation with probiotics tremendously improved the respiratory infections in more than 8000 preterm infants [54]. Several studies have already demonstrated that probiotic supplements can prevent antibiotic-associated diarrhea and infections in the gastrointestinal tract, but also infections at other sites, including sepsis and RTIs [50,87–94]. Table 1 illustrates the relevant pre-clinical and clinical data supporting the use of probiotics against viral diseases.

This supporting evidence strongly supports probiotics' role in modulating the host immune system, suggesting a potential role for probiotics against viral infections. Supplementation involving probiotics could significantly curtail the extremity of SARS-CoV-2 viruses that causes high morbidity and mortality. In addition, probiotics can be an attractive adjunct, as they can impede cytokine storm by invigorating the innate immunity and evading the exaggeration of adaptive immunity; inventing effective therapy will transform the impact of the pandemic on lives as well as economies across the globe. Therefore, supplementation of probiotics in high-risk and severely ill patients, and frontline health workers, might limit the infection and flatten the COVID-19 curve.

**Table 1.** Pre-clinical and clinical data supporting the use of probiotics against viral diseases.

| Probiotics Strains | Against Diseases and Viral Infections | Clinically Tested on | Results from the Clinical Studies | Ref. |
|---|---|---|---|---|
| *B.infantis 35624* | Inflammatory-bowel diseases | Clinical trial on 192 participants | Significant reduction in C reactive protein levels and proinflammatory markers (TNF-$\alpha$ and IL-6) | [9] |
| *B. bifidum* | Influenza virus—(H1N1) | Female mice | Elevated survival rate along with the induction of both humoral and cellular immune responses | [9] |
| *B. lactis* | RTI | Clinical trial on 109 participants | Neonates receiving probiotics had a lower (65%) incidence of respiratory infections as compared to 94% of infants in the control group | [9] |
| *Bacillus subtilis*3 | Influenza virus(H1N1) | Mice | Reduced viralload in lungs and improved survival rate of infected mice | [9] |
| *L. pentosus* | Influenza virus(H1N1) | Female mice | Higher survivalrate and lower viralload in lungs alongwith increased NK cellactivity along with a high expression of IL-12 and IFN-$\alpha$ in the lung | [95] |
| *L. rhamnosus GG and L. gasseri* | Influenza virus(H1N1) | Female mice | Improved clinical symptoms and lower virus load in the lungs | [96] |
| *L. pentosus* | Influenza virus(H1N1) | Female mice | Alleviate survival rate and decreased virus load in the lungs along with increased production of IgA and IgG in bronchoalveolar lavage fluid and plasma | [97] |
| *L. rhamnosus* | Influenza virus(H1N1) | Female mice | Alleviate survival rate with increased secretory IgA production and reduced the expression levels of TNF-$\alpha$ and IL-6 | [98] |
| *L. reuteri* | Inflammatory-bowel diseases | Clinical trial on 40 participants | Useful in improving mucosal inflammation along with increased cytokine expression level of IL-10 and decreased levels of TNF-$\alpha$, IL-1$\beta$ and IL-8 | [99] |

**Table 1.** *Cont.*

| Probiotics Strains | Against Diseases and Viral Infections | Clinically Tested on | Results from the Clinical Studies | Ref. |
|---|---|---|---|---|
| *L. plantarum* | Influenza virus A/PR/8/34 (H1N1) | Female mice | decreased weight loss, increased clinical symptoms and reduced virus load in the lungs of infected mice | [9] |
| *L. plantarum* | Influenza virusH3N2 | Madin–Darby canine kidney cells | Inhibited viral infectivity and proliferation successfully | [100] |
| *L. acidophilus* | Influenza virus(H1N1) | Female mice | Increased expression of antiviral cytokines and chemokines with prevented weight lossand reduced viral load in the lungs | [101] |
| *L. rhamnosus and B. lactis* | Upper-respiratory tract infection | Clinical trial on 231candidates | Lower severity in the probiotics group | [102] |
| *L. rhamnosus* | Influenza virus (H1N1) andrespiratory syncytial virus | Male mice | Decreased risk of lung injury | [103] |
| *L. gasseri* | Influenza virus(H1N1) | Male mice | Reduced expression of IL-6 in the lung tissue and decreased virus load | [104] |
| *L. casei* | Antibiotic-associated diarrhea | Clinical trial on 258 candidates | Effective in the treatment of antibiotic-associated diarrhea in adults and infants | [105] |
| *B. longum, L. rhamnosus, and L. plantarum* | Ventilator-associated pneumonia | Clinical trials on 1083 candidates | Revealed the beneficial role of probiotic strains in reducing the risk of ventilator-associated pneumoniain patients | [106] |
| *L. reuteri Protectis* | Coxsackie-viruses and enterovirus | Human rhabdomyosarcoma and Caco-2 cell lines | Revealed antiviral activity Coxsackievirus and Enterovirus | [107] |
| *L. rhamnosus* | Influenza virus(H1N1) | Female mice | Increased production of IFN-$\gamma$, IL-2and IgA; the increased survival rate and lower viral titer in lungs of infected mice | [108] |
| *Streptococcus thermophilus, L. acidophilus, L. rhamnosus 1, and B. lactis Bb-12.* | Upper-respiratory tract infection | Clinical trials on 6269 participants | Decrease in the prevalence of respiratory tract infections along with the improved quality of life | [109] |
| *Enterococcus faecalis* | Influenza virus and enterovirus | Male mice | Low viral load and improved survival rate | [110] |
| *L. salivarius, L. reuteri, and L. acidophilus* | Influenza virus(H4N6) | Madin–Darby canine kidney cells | Improved expression of IL-1$\beta$, IFN-$\gamma$and IFN-$\alpha$ resulted in protective responses against infection | [65] |
| *L. casei* | Influenza virus(H3N2) | Female mice | Prevented weight loss and higher survival rate | [111] |
| *L. paracasei* | Upper respiratory tract infection | Clinical trial on 233 candidates | Reduced provenance | [112] |
| *L. casei* | Upper respiratory tract infection | Clinical trial on 96 female candidates | Lower incidence of respiratory infections | [113] |
| *L. plantarum* | Influenza H1N1 andH3N2 | Female mice | Significantly lower viral proliferation and increased survival rate | [9] |
| *L. fermentum, L. casei, and L. paracase.* | Upper respiratory tract infection | Clinical trial on 136 patients | 50–60% reduced prevalence of common cold and increased levels of IFN-$\gamma$ andIgA | [51] |
| *B. infantis, L. reuteri, and L. rhamnosus GG* | Multiple diseases | Meta-analysis trials | Probiotics wereeffectivein combating necrotizing enterocolitis, infant colic, antibiotic-associated diarrhea, acute infectious diarrhea and acute respiratory tract infections | [114] |
| *L. gasseri* | Respiratory syncytial virus-A2 strain | Female mice | Eeduced expression of proinflammatory cytokines, with decreased risk of weight loss and lower viral load in the lungs | [115] |
| *B. lactis Bb-12L. rhamnosus GG, L. casei* | Acute otitis andacute respiratory tract infections | - | Reduction in the prevalence of common acute infections and antibiotics utilization | [116] |

## 6. Probiotics and COVID-19: Current Perspectives

The affirmative effects of probiotics species on the ACE receptor are well-stated by Robles-Vera et al., focusing on the anti-hypertensive effects of probiotics [117]. During the period of fermentation of food, probiotics induce the production of significant bioactive peptides with the ability to reduce the activity of ACE enzymes by impeding the active sites [118,119]. Most importantly, the left-over of the dead probiotics also played as promising ACE inhibitors [120]. From the above findings, it can be derived that probiotics could be a promising inhibiter to the ACE receptor that plays a major role as an entry for SARS-CoV-2 to infect the GIT. The notion of utilizing medicines for obstruction of the ACE receptors as a treatment approach to combat COVID-19 was proposed by Fernández-Fernández [121], regardless of the different opinion expressed by Esler and Esler [122]. Additionally, Imai and co-authors [123] have explained the affirmative impact of utilizing an ACE2-blocker to diminish respiratory-distress-syndrome. Notably, prebiotics might also have a significant impact onCOVID-19 by improving the survivability and growth of probiotics. In 2018, Yeh et al. [124] meticulously reviewed twelve studies that scrutinized the effect of probiotic and prebiotic supplements on the infections caused by influenza. Further, the authors concluded that the probiotics and probiotic supplements can enhance the hemagglutination inhibition antibody titers following the vaccination against influenza.

SARS-CoV-2 is a novel emerging virus without any effective therapeutics. Moreover, no research has claimed the promising role of probiotics and prebiotics in preventing/treating COVID-19. Additionally, various registered clinical trials that endeavor to explore the effectiveness of probiotics in treating COVID-19 patients are still ongoing [125]. Most importantly, a number of patients infected with COVID-19 showed dysbiosis in intestinal microbiota underpinning lower amounts of probiotic species, including *Bifidobacterium* and *Lactobacillus* [72], indicating weak immunity of COVID-19-infected patients; thus, patients necessitate nutritional maintenance as well as probiotic/prebiotic supplements to maintain the intestinal flora equilibrium and reduce the chance of infection [126]. As humans have not acquired immunity against the novel COVID-19disease, and the dietary balance at GI microbiota levels is highly essential, a balanced diet involving probiotics-containing foods and immunity-enhancing micronutrients viz., polyphenols; vitamins A, C, and D; and minerals (mainly selenium and zinc) can be highly effective to ease the risk of COVID-19 infection [127]. Early research suggested that the utilization of fermented milk, including probiotics strains, considerably reduced the occurrence of upper respiratory tract infections among elderly adults, children and healthy infants [112,128–130].

The existence of probiotics can help to enhance the anti-microbial peptide production, enhance the attachment of mucins, decrease the pathogenic agent from the mucosal layer, stimulate immunomodulatory agent, ACE inhibitor peptide, anti cholesterolemic, enhance the production of lactoferrin, synthesize ca+ binding protein, maintain the pH, help in neutralizing most of the neurotoxins, etc. (Figure 1). Hence, there is a need to have the probiotics to boost the natural immunity [130,131].

Evidently, based on the aforementioned studies of the impending purpose of probiotics, supplementation involving probiotic bacterial species can be a suitable strategy for treating and inhibiting various viral infections. These interpretations assist the management of probiotics for patients infected with COVID-19. In spite of the absence of any strong evidence supporting these treatments, enhancing the natural immunity of the population using probiotics before, during or after COVID-19 infection is the foremost priority (Figure 5).

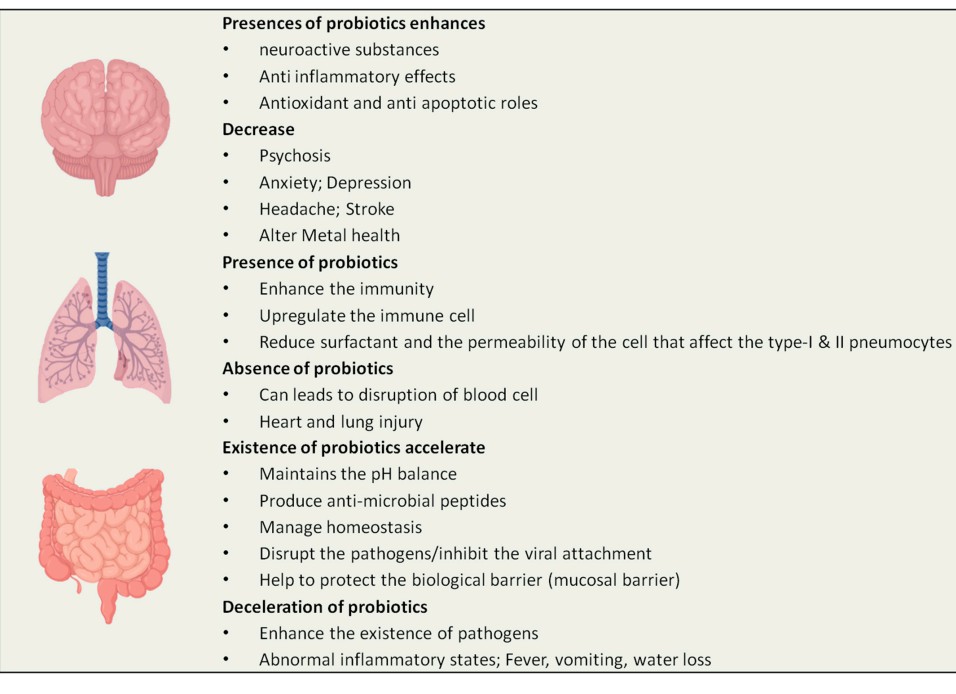

**Figure 5.** Effects of probiotics on different organs of human body.

## 7. Conclusions

Researchers are still in the early stage of understanding the mechanism of SARS-CoV-2 infection in the human body. Evidence is gathering suggesting the benefits of probiotics in regulating the immune system, inhibiting the cytokine storm and boosting adaptive immunity. However, evidence shows that the appropriate usage of current probiotics is safe, even for critically ill and immune-compromised patients [132–135]. Therefore, a clear understanding of the mechanism of probiotics and their mode of use should be determined on an individual basis. In addition, clinical trials, along with biochemical profiling of SARS protein E, are essential before assigning a probiotic in the prophylaxis of COVID-19. When used with caution, probiotic supplementation could reduce the severity of COVID-19 morbidity and mortality. The current situation demands creating awareness among the people about the health benefits of probiotics through social networks at the district, national and international levels to control the spread of COVID-19 infection.

**Author Contributions:** S.S. and M.S., literature search and data analysis: S.p.D. and S.S.; writing—original draft preparation: S.S. and S.M.; writing—review and editing: N.M. and M.S. All authors have read and agreed to the published version of the manuscript.

**Funding:** The authors declare that no funds, grants or other support was received during the preparation of this manuscript.

**Institutional Review Board Statement:** Not applicable.

**Informed Consent Statement:** Not applicable.

**Data Availability Statement:** Not applicable.

**Conflicts of Interest:** The authors declare no conflict of interest.

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
