# Peer review of "Effect of Probiotics on Host-Microbial Crosstalk: A Review on Strategies to Combat Diversified Strain of Coronavirus"

_encyclopedia, doi:10.3390/encyclopedia2020076_

Round 1

Reviewer 1 Report

General comments

The manuscript provides an overview about the potential benefits of using probiotics and prebiotics in managing COVID-19 symptoms.

In general, the work is sufficiently appropriate and quite-well written, anyway I suggest to consider  the manuscript suitable for publication after minor revisions.

Please carefully proof-read spell check the entire manuscript to eliminate grammatical errors throughout the manuscript. Check microorganisms’ names that they must be written in italic.

Few suggestions/critical advice are reported below:

Title: Correct the word “Coronavirus”

Abstract - Last line: please correct and replace the word “improvising” with “improving”

Overview paragraph – Line 19: please use the word “currently” at the beginning of sentences

Overview paragraph – Line 50: check the sentence “the significance of pro-biotics IN PREVENTING viral and bacterial infections…”

Discussion section: Please, mention the new nomenclature of genus Lactobacilllus that was updated in 2020 by Zheng J., Wittouck S., Salvetti E. et al., (2020). A taxonomic note on the genus Lactobacillus: Description of 23 novel genera, emended description of the genus Lactobacillus Beijerink 1901, and union of Lactobacillaceae and Leuconostocaceae. https://doi.org/10.1099/ijsem.0.004107 and update microorganisms names with the new nomenclature.

The authors can easily use this tool to check new species names: http://lactotax.embl.de/wuyts/lactotax/

Also check that all microorganisms names are written properly. They should be written in italic

Page 4 – last two lines: please check English grammar errors “to DECREASE viral load in the lungs, TO EASE clinical symptoms, and TO IMPROVE animal health, and survival rates”

PAGE 5 – LINE 2: Please check plural/singular grammar errors “probiotics also COLONIZE at distant mucosal sites, including the lung; and also ENHANCE the systemic immune responses

PAGE 5 – LINE 4: I suggest to rewrite interleukins: …such as IL10, IL12, IL17

PAGE 5 – LINE 5: “Some immunomodulatory AGENTS” and please rephrase the entire sentence

Figure 3: I suggest to use the term “microbiota or microbiome” instead of microflora

“COVID-19 affecting the Gut-lung axis crosstalk” paragraph: The GIT and lung are amongst the sections in human body that harbors THE MAJORITY OF HUMAN microbiota

Figure 4: I suggest to use an image with higher resolution

“Supporting evidence of usage of probiotics to combat COVID-19” paragraph: please, carefully check singular/plural English grammar errors within the text

“Supporting evidence of usage of probiotics to combat COVID-19” paragraph: please check this sentence: “demonstrated enteral supplementation with probiotics improved tremendously from the respiratory infections” it might be better to write “demonstrating that enteral supplementation with probiotics improved tremendously the respiratory infections”

“So probiotics supplements ARE strongly recommended for THEIR involvement IN the gut during COVID-19” please rephrase also the last part of the sentence to better clarify the meaning

“Probiotics and COVID-19: Current Perspectives” paragraph: probiotics-species

Please rephrase the sentence “At the period of fermentation of food” with “During food fermentation”

Correct probiotics INDUCE

Correct TO REDUCE

Please, carefully proof-read spell-chcek the entire paragraph for English grammar errors

Article Highlights section:

Patients infected with SARS-CoV-2 virus exhibited intestinal and microbial dysbiosis with a reduced BENEFICIAL MICROBES POPULATION

Please, carefully proof-read spell-check the entire paragraph for English grammar errors

Author Response

Responses to reviewer 1

Dear Reviewer,

We very much appreciated the encouraging, critical and constructive comments on this manuscript by the reviewer. The comments have been very thorough and useful in improving the manuscript. We are submitting the revised manuscript with the suggestion incorporated. The manuscript has been revised as per the comments given by the reviewer, and our responses to all the comments are as follows:

The manuscript provides an overview about the potential benefits of using probiotics and prebiotics in managing COVID-19 symptoms. In general, the work is sufficiently appropriate and quite-well written, anyway I suggest to consider the manuscript suitable for publication after minor revisions. Please carefully proof-read spell check the entire manuscript to eliminate grammatical errors throughout the manuscript. Check microorganisms’ names that they must be written in italic. Few suggestions/critical advice are reported below:

Point 1: Title: Correct the word “Coronavirus”

Author response: Thank you so much for your minute observation. We have incorporated the change in the revised manuscript.

Point 2: Abstract - Last line: please correct and replace the word “improvising” with “improving”

Author response: Thank you so much for your suggestion. Yes, we have restructured the sentence for better understanding. Hopefully you will find it logical.

Point 3: Overview paragraph – Line 19: please use the word “currently” at the beginning of sentences

Author response: We agree with this and have incorporated your suggestion.

Point 4: Overview paragraph – Line 50: check the sentence “the significance of pro-biotics IN PREVENTING viral and bacterial infections…”

Author response: The suggested correction has been made. The sentence now reads:

“Moreover, few clinical evidence also illustrated the significance of probiotics in preventing viral and bacterial infections, including RTIs, sepsis, and gastroenteritis”

Point 5: Discussion section: Please, mention the new nomenclature of genus Lactobacilllus that was updated in 2020 by Zheng J., Wittouck S., Salvetti E. et al., (2020). A taxonomic note on the genus Lactobacillus: Description of 23 novel genera, emended description of the genus Lactobacillus Beijerink 1901, and union of Lactobacillaceae and Leuconostocaceae. https://doi.org/10.1099/ijsem.0.004107 and update microorganisms names with the new nomenclature. The authors can easily use this tool to check new species names: http://lactotax.embl.de/wuyts/lactotax/. Also check that all microorganisms names are written properly. They should be written in italic

Author response: As suggested by the reviewer, the manuscript has been revised thoroughly with the new nomenclatures according to :

  1. A taxonomic note on the genus Lactobacillus: Description of 23 novel genera, emended description of the genus LactobacillusBeijerink 1901, and union of Lactobacillaceae and Leuconostocaceae.
  2. Lactotax tool (http://lactotax.embl.de/wuyts/lactotax/).

Point 6: Page 4 – last two lines: please check English grammar errors “to DECREASE viral load in the lungs, TO EASE clinical symptoms, and TO IMPROVE animal health, and survival rates”

Author response: Thank you for pointing this out. As suggested, the above statement has been correctly rewritten in the revised manuscript.

Point 7: PAGE 5 – LINE 2: Please check plural/singular grammar errors “probiotics also COLONIZE at distant mucosal sites, including the lung; and also ENHANCE the systemic immune responses

Author response: As suggested, to avoid confusion the above statement has been correctly rewritten in the revised manuscript.

Point 8: PAGE 5 – LINE 4: I suggest to rewrite interleukins: …such as IL10, IL12, IL17                                                                                                 PAGE 5 – LINE 5: “Some immunomodulatory AGENTS” and please rephrase the entire sentence

Author response: We appreciate this recommendation, the manuscript has been meticulously checked and substantively proof read to remove any typographical or vocabulary error and all the typo mistakes as pointed have been now corrected in the revised manuscript.

Point 9: Figure 3: I suggest to use the term “microbiota or microbiome” instead of microflora

Author response: The necessary modifications have been incorporated in the illustration of figure 3.

Point 10: “COVID-19 affecting the Gut-lung axis crosstalk” paragraph: The GIT and lung are amongst the sections in human body that harbors THE MAJORITY OF HUMAN microbiota

Author response: The change has been incorporated in the text accordingly.

Point 11: Figure 4: I suggest to use an image with higher resolution

Author response: Figures 4 are modified accordingly.

Point 12: “Supporting evidence of usage of probiotics to combat COVID-19” paragraph: please, carefully check singular/plural English grammar errors within the text                                                                                        “Supporting evidence of usage of probiotics to combat COVID-19” paragraph: please check this sentence: “demonstrated enteral supplementation with probiotics improved tremendously from the respiratory infections” it might be better to write “demonstrating that enteral supplementation with probiotics improved tremendously the respiratory infections”                                                                                            “So probiotics supplements ARE strongly recommended for THEIR involvement IN the gut during COVID-19” please rephrase also the last part of the sentence to better clarify the meaning                                 “Probiotics and COVID-19: Current Perspectives” paragraph: probiotics-species                                                                                                         Please rephrase the sentence “At the period of fermentation of food” with “During food fermentation”                                                               Correct probiotics INDUCE  -> Correct TO REDUCE                             Please, carefully proof-read spell-chcek the entire paragraph for English grammar errors

Author response: The grammatical and typo errors are rectified in the whole text and modified accordingly. The suggested change has been included in subsection “Supporting evidence of usage of probiotics to combat COVID-19”.

Point 13: Article Highlights section:

Patients infected with SARS-CoV-2 virus exhibited intestinal and microbial dysbiosis with a reduced BENEFICIAL MICROBES POPULATION

Please, carefully proof-read spell-check the entire paragraph for English grammar errors

Author response: Agreed.  As suggested, few mismatches were thoroughly rechecked and rewritten in the revised manuscript.

Reviewer 2 Report

Although this review sounds unique, immunity thereby probiotics is not the only factor that controls Covid. Nothing yet is confirmed about probiotics and Covid directly. Authors should explain more on that point.

Author Response

Responses to reviewer 2

Dear Reviewer,

We very much appreciated the encouraging, critical and constructive comments on this manuscript by the reviewer. The comments have been very thorough and useful in improving the manuscript. We are submitting the revised manuscript with the suggestion incorporated. The manuscript has been revised as per the comments given by the reviewer, and our responses to all the comments are as follows:

Point 1: Although this review sounds unique, immunity thereby probiotics is not the only factor that controls Covid. Nothing yet is confirmed about probiotics and Covid directly. Authors should explain more on that point.

Author response: In light of the reviewer’s comment: we have included a paragraph in the section Supporting evidence of usage of probiotics to combat COVID-19 “These supporting evidences strongly supports probiotics’ role in modulating the host immune system, suggesting a potent role for probiotics against viral infections. Supplementation involving probiotics could significantly curtail the extremity of SARS-CoV 2 viruses that causes high morbidity and mortality. In addition probiotics can be an attractive adjunct, as it can impede cytokine storm by invigorating the innate immunity and evading the exaggeration of adaptive immunity, Inventing effective therapy will transform the impact of the pandemic on lives as well as economies across the globe. Therefore, supplementation of probiotics in high risk and severely ill patients, and frontline health workers, might limit the infection and flatten the COVID-19 curve.” in the revised manuscript which provides information about probiotics that can be used as Weapon to combat COVID-19.

Reviewer 3 Report

This paper deals with the role of probiotics in relation to COVID-19.

The paper might be eligible for publication, after major revisions have been made, and after the corrected version is resubmitted for peer-review.

Most important issue with this paper that it displays the role of probiotics to prevent / treat  COVID-19 way too optimistic without any sound scientific justification.

Other remarks:

English: should be thoroughly checked and corrected throughout the article.

Abstract: 

- "This review aims to sketch out the prospective role of probiotics and prebiotics on improvising the standard of health in common people." --> improvising health? What do you mean?

Overview:

- Please discuss in the section "overview", in relation to "involving social distancing, mask-wearing, personal hygiene, quarantines, and lockdowns" also the downside of these measures, i.e. a diminished exposure to beneficial microbes, by referring to references: 
    - Larsen, Olaf FA, and Linda HM van de Burgwal. "On the Verge of a Catastrophic Collapse? The Need for a Multi-Ecosystem Approach to Microbiome Studies." Frontiers in Microbiology 12 (2021).
    - Finlay, B. Brett, et al. "The hygiene hypothesis, the COVID pandemic, and consequences for the human microbiome." Proceedings of the National Academy of Sciences 118.6 (2021).

- When discussing the relation COVID-19 and gut, I think it is essential to at least include the following reference that shows that the status of the gut microbiota is a predictor for disease severity:
    - Yeoh, Yun Kit, et al. "Gut microbiota composition reflects disease severity and dysfunctional immune responses in patients with COVID-19." Gut 70.4 (2021): 698-706.

- Figure 1 displays a rather detailed representation of the virus with all kinds of sub-compartments. The question is what this information adds to the role of probiotics that is the topic of discussion in this paper.

Discussion:

- Please use the WHO definition reinforced by ISAPP for probiotics:
    - Hill, Colin, et al. "Expert consensus document: The International Scientific Association for Probiotics and Prebiotics consensus statement on the scope and appropriate use of the term probiotic." Nature reviews Gastroenterology & hepatology (2014).

- "Probiotics serve enormous metabolic advantages and took the immunity to next level. immunity (by enhancing specific and non-specific immune system) [46]"--> sentence not correct

- "Probiotics belonging to the genera Bifidobacterium and Lactobacillus have been commonly utilized for their wide-variety of benefits to the health by prevention and treatment of many pathogenic infections" --> this is, unfortunately, a way too optimistic view on probiotics yet.

- Figure 3: this picture suggests that ingestion of probiotics fully protects against gut and lung dysbiosis caused by SARS-CoV-2, which is currently  premature to claim.

Supporting evidence of usage of probiotics to combat COVID-19:

- "Despite several probable medications to treat newly emerging SARS-CoV-2, it has been observed that with intake of optimized probiotics supplement most of the people are withstanding COVID-19, due to booted immunity. Implication of probiotics strain, specifically Bifidobacteria and Lactobacilli,
uplifted the health benefits and a significant stimuation towards recovery [82]."--> This is an extremely bold claim that cannot be made based on the existing data yet. Moreover: reference 82 deals with probiotics that are currently on the market and and may be of use for indications that are commonly also found during a SARS-VoV2 infection, suggesting that they may be of use to flatten the curve , but are not being tested for COVID-19.

Conclusions:

- "However, evidences also support the idea of using probiotics with caution, especially for critically ill and immune-compromised patients."
    The safety profile of probiotics has been nicely described in the following papers, showing that probiotics, when properly used, are safe (so also for immune compromised patients). Please refer to these papers:
        - Van den Nieuwboer, M., et al. "The administration of probiotics and synbiotics in immune compromised adults: is it safe?." Beneficial microbes 6.1 (2015): 3-17.
        - Van den Nieuwboer, M., et al. "Safety of probiotics and synbiotics in children under 18 years of age." Beneficial microbes 6.5 (2015): 615-630.
        - Van den Nieuwboer, M., et al. "Probiotic and synbiotic safety in infants under two years of age." Beneficial microbes 5.1 (2014): 45-60.
        - Larsen, O. F. A., et al. "Probiotics for healthy ageing: innovation barriers and opportunities for bowel habit improvement in nursing homes." Agro Food Industry Hi Tech 28.5 (2017): 12-15.

Author Response

Responses to reviewer 3

Dear Reviewer,

We very much appreciated the encouraging, critical and constructive comments on this manuscript by the reviewer. The comments have been very thorough and useful in improving the manuscript. We are submitting the revised manuscript with the suggestion incorporated. The manuscript has been revised as per the comments given by the reviewer, and our responses to all the comments are as follows:

Point 1: This paper deals with the role of probiotics in relation to COVID-19. The paper might be eligible for publication, after major revisions have been made, and after the corrected version is resubmitted for peer-review. Most important issue with this paper that it displays the role of probiotics to prevent / treat  COVID-19 way too optimistic without any sound scientific justification.

Author response: Thank you for giving me the opportunity to submit a revised draft of my manuscript titled “Effect of probiotics on host-microbial crosstalk: a systemic review on strategies to combat diversified strain of Coronavirus” to Encyclopedia, MDPI.

Given the severe infection, poor prognosis, and the low number of available effective drugs, potential prevention and treatment strategies for COVID-19 need to be urgently developed. Most importantly, probiotics as nutritional supplements are long practiced in different cuisines across various countries, the emerging scientific evidence supports the antiviral and general immune-strengthening health effects of the probiotics. Keeping all these information in mind, repurposing the usage of natural compounds such as probiotics and prebiotics can be an effectual therapeutic approach in blocking and/or reducing SARS-CoV-2 severity. In this regard, we have described the existing curative and preventive trial studies focused on the usage of probiotics and prebiotics to combat viral infections, in this manuscript. Moreover, the possible application of probiotic bacteria as a prophylactic approach against COVID-19 has also been outlined in the present study. 

Point 2: Other remarks: English: should be thoroughly checked and corrected throughout the article.

Author response: The grammatical and typo errors are rectified in the whole text and modified accordingly.

Point 3: Abstract: 

- "This review aims to sketch out the prospective role of probiotics and prebiotics on improvising the standard of health in common people." --> improvising health? What do you mean?

Author response: Thank you so much for your comments. Abstract section is thoroughly revised considering your valuable suggestion.

Point 4: Overview:

- Please discuss in the section "overview", in relation to "involving social distancing, mask-wearing, personal hygiene, quarantines, and lockdowns" also the downside of these measures, i.e. a diminished exposure to beneficial microbes, by referring to references: 
    - Larsen, Olaf FA, and Linda HM van de Burgwal. "On the Verge of a Catastrophic Collapse? The Need for a Multi-Ecosystem Approach to Microbiome Studies." Frontiers in Microbiology 12 (2021).
    - Finlay, B. Brett, et al. "The hygiene hypothesis, the COVID pandemic, and consequences for the human microbiome." Proceedings of the National Academy of Sciences 118.6 (2021).

Author response: Thank you for bringing this point to our attention. We have included few lines in the revised manuscript under the section Overview that explains the downside of social distancing, mask-wearing, personal hygiene, quarantines, and lockdowns on beneficial microbes associated to human. The suggested references on Microbiome Studies were consulted and duly cited in the relevant places in the revised manuscript.

Point 5: - When discussing the relation COVID-19 and gut, I think it is essential to at least include the following reference that shows that the status of the gut microbiota is a predictor for disease severity:
    - Yeoh, Yun Kit, et al. "Gut microbiota composition reflects disease severity and dysfunctional immune responses in patients with COVID-19." Gut 70.4 (2021): 698-706.

Author response: Thank you so much for your comment. The suggested references on relationship of COVID-19 and gut is included and cited in the relevant places in the revised manuscript.

Point 6: - Figure 1 displays a rather detailed representation of the virus with all kinds of sub-compartments. The question is what this information adds to the role of probiotics that is the topic of discussion in this paper.

Author response: Thank you for bringing this point to our attention. Knowledge of the detailed structure of virus particles is an essential prerequisite to our understanding of many aspects of virology. In addition to Lungs, it is reported that angiotensin converting enzyme 2 (ACE2) receptor is also identified in GIT, and direct colonization of the gut ACE2 receptors through the ingestion of the virus is probably liable for the GIT symptoms in connection with COVID-19. Depending on the Cryo-EM structural investigation of S, it was revealed that the main reason for the rapid spread of SARS-CoV-2 is the S protein, having a 10-20times higher affinity to ACE2. Most importantly, Imai and co-authors [113] have explained the affirmative impact of utilizing an ACE2-blocker to diminish respiratory-distress-syndrome. Keeping this in mind, we have included Figure 1 “The schematic structure of SARS-CoV-2. The viral surface proteins, spike, envelope and membrane proteins. The single-stranded positive sense viral RNA is associated with the nucleocapsid protein”.

Point 7: Discussion:

- Please use the WHO definition reinforced by ISAPP for probiotics:
    - Hill, Colin, et al. "Expert consensus document: The International Scientific Association for Probiotics and Prebiotics consensus statement on the scope and appropriate use of the term probiotic." Nature reviews Gastroenterology & hepatology (2014).

Author response: We very much appreciate reviewer’s suggestion and we have updated WHO definition reinforced by ISAPP for probiotics.

Point 8: - "Probiotics serve enormous metabolic advantages and took the immunity to next level. immunity (by enhancing specific and non-specific immune system) [46]"--> sentence not correct

Author response: We thank the reviewer for his appreciation and critical suggestions. We have corrected these spelling errors. Furthermore, the manuscript has been substantively proof read to remove any such vocabulary or typographical mistakes.

Point 9: - "Probiotics belonging to the genera Bifidobacterium and Lactobacillus have been commonly utilized for their wide-variety of benefits to the health by prevention and treatment of many pathogenic infections" --> this is, unfortunately, a way too optimistic view on probiotics yet.

Author response: As suggested by the reviewer, to avoid confusion the above statement has been correctly rewritten in the revised manuscript.

Point 10: - Figure 3: this picture suggests that ingestion of probiotics fully protects against gut and lung dysbiosis caused by SARS-CoV-2, which is currently  premature to claim.

Author response: Thank you for pointing this out. Figure 3 with caption “Schematic representation of the bidirectional cross-talk between the Gut-Lung axis” provides a general idea that suggests that probiotics can be important factor  in improving the gut and lung dysbiosis condition caused by SARS-CoV-2.

Point 11: Supporting evidence of usage of probiotics to combat COVID-19:

- "Despite several probable medications to treat newly emerging SARS-CoV-2, it has been observed that with intake of optimized probiotics supplement most of the people are withstanding COVID-19, due to booted immunity. Implication of probiotics strain, specifically Bifidobacteria and Lactobacilli,
uplifted the health benefits and a significant stimuation towards recovery [82]."--> This is an extremely bold claim that cannot be made based on the existing data yet. Moreover: reference 82 deals with probiotics that are currently on the market and and may be of use for indications that are commonly also found during a SARS-VoV2 infection, suggesting that they may be of use to flatten the curve , but are not being tested for COVID-19.

Author response: We agree with the reviewer’s assessment. Accordingly, the section “Supporting evidence of usage of probiotics to combat COVID-19” has been thoroughly revised. Furthermore, the manuscript has been substantively proof read to remove any such vocabulary or typographical mistakes.

Point 12: Conclusions:

- "However, evidences also support the idea of using probiotics with caution, especially for critically ill and immune-compromised patients."
    The safety profile of probiotics has been nicely described in the following papers, showing that probiotics, when properly used, are safe (so also for immune compromised patients). Please refer to these papers:
        - Van den Nieuwboer, M., et al. "The administration of probiotics and synbiotics in immune compromised adults: is it safe?." Beneficial microbes 6.1 (2015): 3-17.
        - Van den Nieuwboer, M., et al. "Safety of probiotics and synbiotics in children under 18 years of age." Beneficial microbes 6.5 (2015): 615-630.
        - Van den Nieuwboer, M., et al. "Probiotic and synbiotic safety in infants under two years of age." Beneficial microbes 5.1 (2014): 45-60.
        - Larsen, O. F. A., et al. "Probiotics for healthy ageing: innovation barriers and opportunities for bowel habit improvement in nursing homes." Agro Food Industry Hi Tech 28.5 (2017): 12-15.

Author response: As suggested, in the revised chapter, the aforementioned references have been incorporated in relevant place and also highlighted through track changes in the reference section.

Round 2

Reviewer 1 Report

The manuscript have been sufficiently improved and it can be considered suitable for publication in the present form.

Author Response

We appreciate the positive feedback from the reviewer.  

Reviewer 2 Report

Thanks for the authors for addressing my comments.

Author Response

(The authors gave the same response as above.)

Reviewer 3 Report

This manuscript can be published provided the following corrections are being implemented:

Page 12:

“As existence of probiotics can help to enhance the anti-microbial peptide production, enhance the attachment of mucins, decrease the pathogenic agent from the mucosal layer, stimulate immunomodulatory agent, ACE inhibitor peptide, anti cholesterolemic, en-hance production of lactoferrin, synthesize ca+ binding protein, maintains the pH, helps in neutralizing most of neurotoxins etc fig.1. Hence it’s the need of the hour to have the probiotics to boot of the natural immunity [121-124]”

à Please remove reference 124 here because it has nothing to do with it.

Conclusion:

However, evidences also support the idea of using probiotics with caution, especially for critically ill and immune-compromised patients [123-127].

à No, exactly the opposite: references 123 – 127  show that the appropriate usage of current probiotics is safe, even for critically ill and  immune compromised patients. Please correct this accordingly.

References:

Please put a hard return between reference 128 and reference 129 to separate them from each other.

Author Response

Responses to reviewer 3

Dear Reviewer,

We very much appreciated the encouraging, critical and constructive comments on this manuscript by the reviewer. The comments have been very thorough and useful in improving the manuscript. We are submitting the revised manuscript with the suggestion incorporated. The manuscript has been revised as per the comments given by the reviewer, and our responses to all the comments are as follows:

Point 1:- Page 12:“As existence of probiotics can help to enhance the anti-microbial peptide production, enhance the attachment of mucins, decrease the pathogenic agent from the mucosal layer, stimulate immunomodulatory agent, ACE inhibitor peptide, anti cholesterolemic, enhance production of lactoferrin, synthesize ca+ binding protein, maintains the pH, helps in neutralizing most of neurotoxins etc fig.1. Hence it’s the need of the hour to have the probiotics to boot of the natural immunity [121-124]”         à Please remove reference 124 here because it has nothing to do with it.

Author response: Thank you so much for your comments. As suggested, the reference 124 has been removed.

Point 2:- Conclusion: However, evidences also support the idea of using probiotics with caution, especially for critically ill and immune-compromised patients [123-127].                                                                             à No, exactly the opposite: references 123 – 127  show that the appropriate usage of current probiotics is safe, even for critically ill and  immune compromised patients. Please correct this accordingly.

 Author response: Thank you so much for your minute observation. We have incorporated the change in the revised manuscript.

Point 3: References: Please put a hard return between reference 128 and reference 129 to separate them from each other.

Author response: We agree with this and have incorporated your suggestion.

Round 3

Reviewer 3 Report

Can be published as such